# Interventional Causal Representation Learning

**Kartik Ahuja**[*]       **Yixin Wang**[†]       **Divyat Mahajan**[*]       **Yoshua Bengio**[*‡]

## Abstract

The theory of identifiable representation learning aims to build general-purpose methods that extract high-level latent (causal) factors from low-level sensory data. Most existing works focus on identifiable representation learning with observational data, relying on distributional assumptions on latent (causal) factors. However, in practice, we often also have access to interventional data for representation learning, e.g. from robotic manipulation experiments in robotics, from genetic perturbation experiments in genomics, or from electrical stimulation experiments in neuroscience. How can we leverage interventional data to help identify high-level latents? To this end, we explore the role of interventional data for identifiable representation learning in this work. We study the identifiability of latent causal factors with and without interventional data, under minimal distributional assumptions on latents. We prove that, if the true latent maps to the observed high-dimensional data via a polynomial function, then representation learning via minimizing standard reconstruction loss (used in autoencoders) can identify the true latents up to affine transformation. If we further have access to interventional data generated by hard *do* interventions on some latents, then we can identify these intervened latents up to permutation, shift and scaling.

## 1 Introduction

Modern deep learning models like GPT-3 (Brown et al., 2020) and CLIP (Radford et al., 2021) are remarkable representation learners (Bengio et al., 2013). Despite the successes, these models continue to be far from the human ability to adapt to new situations (distribution shifts) or carry out new tasks (Geirhos et al., 2020; Bommasani et al., 2021). Humans encapsulate the causal knowledge of the world in a way that is highly reusable and recomposable (Goyal and Bengio, 2020), which helps them adapt to new tasks in an ever-distribution-shifting world. How to make modern deep learning models extract a similar causal understanding of the world? This question is central to the emerging field of causal representation learning (Schölkopf et al., 2021).

A core task in causal representation learning is *provable representation identification*, i.e. developing conditions under which representation learning algorithms can provably identify latent objects (or factors) and their causal relationships. Towards understanding this task, several notable works have shown that provable representation identification for arbitrary data generation process (DGP) is impossible if we only enforce the independence between the latent factors (Hyvärinen and Pajunen, 1999; Locatello et al., 2019). Yet, real data generation processes often have additional structure we can leverage to achieve provable representation identification. Such structures include the independence between the latent factors conditional on auxiliary information (Khemakhem et al., 2020a), the sparsity of the causal connections among the latents (Lachapelle et al., 2022), and the sparsity of the mechanisms that govern the variation of the latents (Locatello et al., 2020; Ahuja et al., 2022a; Klindt et al., 2020).

---

[*]Mila - Quebec AI Institute, Université de Montréal, Quebec, Canada
[†]University of Michigan, Ann Arbor, Michigan, USA
[‡]CIFAR Senior Fellow and CIFAR AI Chair

36th Conference on Neural Information Processing Systems (NeurIPS 2022).

Despite these efforts toward provable representation identification, most existing works focus on representation learning with observational data, relying on distributional assumptions on latent (causal) factors to achieve identification. However, in practice, we often also have access to interventional data for representation learning, e.g. from robotic manipulation experiments in robotics (Collins et al., 2019), from genetic perturbation experiments in genomics (Dixit et al., 2016), or from electrical stimulation experiments in neuroscience (Nejatbakhsh et al., 2021). In this work, we seek to understand how we can leverage such interventional data to identify high level (causal) factors from low level data. Our key findings are summarized below.

- Under the assumption that the true latent factors map to the high dimensional observations via a finite degree multivariate polynomial, we first show that it is possible to achieve affine identification with respect to the true latents with minimal assumptions on the support of the true latents and no further distributional assumptions.
- If we also observe data where some latents undergo a hard $do$ intervention (Pearl, 2009), then we can guarantee affine identification up to the block of these hard intervened latents. As a result, if in some environments only one latent variable undergoes a $do$ intervention, then those latents are identified up to permutation, shift and scaling.

Due to space limitations, we relegate the related works to the Appendix.

## 2   Representation Identification with Observational & Interventional Data

**The data generating process.**   We begin with defining the data generation process. We observe $x \in \mathcal{X} \subseteq \mathbb{R}^n$ and it is generated from underlying latent variables $z \in \mathcal{Z} \subseteq \mathbb{R}^d$, with $n \geq d$. Suppose the data generating process follows

$$z \sim \mathbb{P}_Z, \quad x \leftarrow g(z), \tag{1}$$

where $\mathbb{P}_Z$ is the distribution from which the latent $z$ is sampled and $x$ is the observed data point that is rendered using an injective decoder $g : \mathbb{R}^d \to \mathbb{R}^n$ from the underlying latent. We denote $\mathcal{Z}$ as the support of $\mathbb{P}_Z$, and $\mathcal{X} = g(\mathcal{Z})$. Hence, the support of observations $x$ is $\mathcal{X}$.

**The identifiable representation learning task.**   To perform representation learning, we learn encoders (also known as the representation function) that aim to estimate the true latents $z$. Specifically, we aim to find an encoder $f : \mathbb{R}^n \to \mathbb{R}^d$ and a decoder $h : \mathbb{R}^d \to \mathbb{R}^n$ such that the encoder and decoder jointly satisfy the following reconstruction identity. For all $x \in \mathcal{X}$ [4]

$$h \circ f(x) = x. \tag{2}$$

We denote the learned representation as $\hat{z} \triangleq f(x)$, where $\hat{z}$ is the latent that the encoder guesses. Note that equation (2) is highly underspecified and can have many solutions that are not guaranteed to identify the latents $z$ (e.g., take any solution $f, h$ of the reconstruction identity, $b \circ f, h \circ b^{-1}$ is another valid solution where $b$ is an invertible map.). In many problems, exact identification of latents is not needed, nor reasonable (for example we do not care about the labels given to each latent, i.e., about coordinate permutations of $z$). Therefore, we study conditions under which the true latents are identifiable up to certain transformations, e.g. affine transformations, coordinate permutations, etc.

**Overview of results.**   Below we present a suite of identifiability results. We first show that, when the true decoder $g$ is a polynomial of $z$ with known degree and the learned decoder $h$ is also a polynomial with the same degree, then latents $z$ can be identified up to affine transfomations using the encoder learned from the reconstruction identity (Equation (2)). We relax the result to settings where we do not know the exact degree of $g$ but we know an upper bound on it. We also provide approximate affine identification guarantees for the setting when $g$ is $\epsilon$-approximated by a polynomial. In the second half, we investigate the role of interventions. If we observe data where some latents undergo a hard $do$ intervention (Pearl, 2009), then we can guarantee affine identification up to the block of these hard intervened latents. As a result, if in some environments only one latent variable variable undergoes a $do$ intervention, then that latent is identifiable up to shift and scaling.

---

[4]The identity requires the reconstruction at all points the support of $X$. We can also extend our results for settings (e.g., $X$ is a continuous random vector) where the identity holds almost everywhere in $\mathcal{X}$.

## 2.1 Affine identifiability with observational data

**Assumption 1.** *The interior of the support of $Z$ denoted as $\mathcal{Z}$ is a non-empty subset of $\mathbb{R}^d$.* [5]

**Assumption 2.** *The decoder $g$ is a polynomial of degree $p$ and its corresponding coefficient matrix (a.k.a. weight matrix) $G$ has full column rank. $g$ is determined by $G$ as follows. For all $z \in \mathbb{R}^d$*

$$g(z) = G[1, z, z\bar{\otimes}z, \cdots, \underbrace{z\bar{\otimes}\cdots\bar{\otimes}z}_{p \text{ times}}]^t \tag{3}$$

*where $\bar{\otimes}$ represents the Kronecker product with all distinct entries, for example, $z = [z_1, z_2]$, then $z\bar{\otimes}z = [z_1^2, z_1 z_2, z_2^2]$.*

In identifiable representation learning, it is common to assume that $g$ is injective or else the problem of identification can become ill defined as multiple latents can give rise to the same observation $x$. If the matrix $G \in \mathbb{R}^{n \times q}$ has full column rank of $q$, then $g$ is guranteed to be injective; the proof of this claim is in the Appendix. The full-column-rank condition for $G$ in Assumption 2 implicitly requires that the dimensionality $n$ of the data is greater than the number of terms in the polynomial of degree $p$, namely $n = \mathcal{O}(d^p)$, where $d$ is the dimensionality of $z$.

**Assumption 3.** *The learned decoder $h$ is a polynomial of degree $p$ and its corresponding coefficient matrix $H$ has a full column rank. $h$ is determined by $H$ as follows. For all $z \in \mathbb{R}^d$*

$$h(z) = H[1, z, z\bar{\otimes}z, \cdots, \underbrace{z\bar{\otimes}\cdots\bar{\otimes}z}_{p \text{ times}}]^t \tag{4}$$

*where $\bar{\otimes}$ represents the Kronecker product with all distinct entries.*

Recall $\hat{z} = f(x)$. Here we can substitute $x = g(z)$ to get $\hat{z} = f \circ g(z) = a(z)$, where $a \triangleq f \circ g$. In the next theorem, we show that $a$ is an affine transform under the assumptions stated above.

**Theorem 1.** *If the data generation follows equation (1) and Assumptions 1-3 hold, then the encoder $f$ and decoder $h$ that solve reconstruction identity in equation (2) achieve affine identification, i.e., $\forall z \in \mathcal{Z}, \hat{z} = Az + c$, where $\hat{z}$ is the output of the encoder $f$ and $z$ is the true latent and $A$ is an invertible $d \times d$ matrix and $c \in \mathbb{R}^d$.*

The proof is provided in the Appendix. To get some intuition for the proof, we consider $z$ to be one-dimensional, $g$ to be a degree two polynomial, and $x$ to be three-dimensional. By solving the reconstruction identity on all $x$, we obtain that $\hat{z}$ is at most a degree-two polynomial of $z$. We also obtain that $\hat{z}^2$ is also a polynomial of at most degree two in $z$. If $\hat{z}$ is a degree-two polynomial of $z$, then $\hat{z}^2$ is degree four, which cannot be the case. Therefore, $\hat{z}$ is a degree-one polynomial.

**Extension of Theorem 1 when degree $p$ is unknown.** In the theorem above, we required that the degree $p$ is known. We can extend the above findings to settings where we know an upper bound on the value $p$, denoted as $s$. Through a simple iterative procedure, which we describe next, we claim the learner can achieve affine identification. The learner first tries to solve the reconstruction identity with with a polynomial $h(\cdot)$ of degree equal to the upper bound $s$. We assume that $H$ is full rank, which implicitly requires $n$ to be sufficiently large, i.e., $n = \mathcal{O}(d^s)$. If $s > p$, then we can show that there is no solution to the reconstruction identity in equation (2) (see Appendix for further justification). In the next step, the learner searches over full rank polynomials $h(\cdot)$ of degree $s - 1$ and this procedure is repeated until the degree is $p$, which is when the reconstruction identity is satisfied.

**Extension beyond polynomial decoders.** From Stone-Weiresstrass theorem (Rudin et al., 1976), we know that a continuous map on a closed and bounded set can be approximated with a high dimensional polynomial. Inspired by this, we extend our results to a class of maps $g(\cdot)$ that are $\epsilon$ approximated by a polynomial of sufficiently high degree. Essentially, we show that the map $a(\cdot)$ that connects $\hat{z}$ to true $z$ is an approximately linear map, i.e., the norm of the weight on higher order terms in the polynomial expansion is sufficiently small.

---

[5]Here we work with $(\mathbb{R}^d, \|\|_2)$ as the metric space. Recall a point is said to be in the interior if there exists an $\epsilon$ ball containing that point that is strictly in the set. The set of all the interior points define the interior of the set.

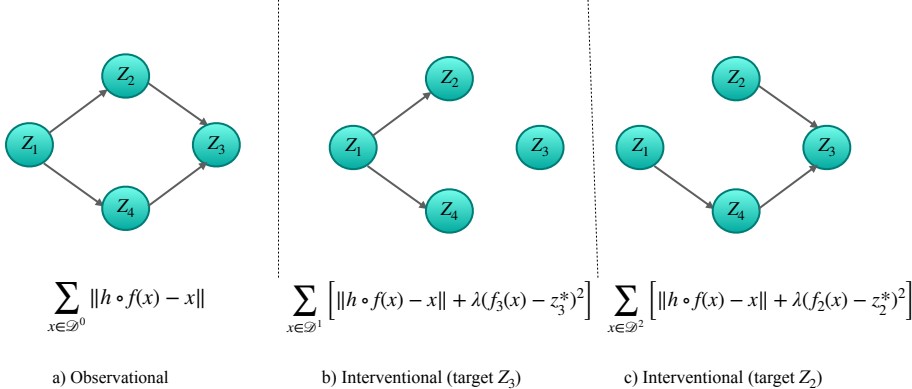

$$\sum_{x \in \mathscr{D}^0} \|h \circ f(x) - x\|$$

$$\sum_{x \in \mathscr{D}^1} \left[ \|h \circ f(x) - x\| + \lambda (f_3(x) - z_3^*)^2 \right]$$

$$\sum_{x \in \mathscr{D}^2} \left[ \|h \circ f(x) - x\| + \lambda (f_2(x) - z_2^*)^2 \right]$$

a) Observational      b) Interventional (target $Z_3$)      c) Interventional (target $Z_2$)

Figure 1: Illustrating the data generation process using a simple SCM. Figure 1a shows the causal DAG and the associated reconstruction loss used on observational data. Figure 1b and c show the intervened causal DAG and associated reconstruction loss and penalty arising from $do$ intervention constraints on the decoder, with $z_3^*$ (resp. $z_2^*$) as the value of the intervention in b (resp. in c).

## 2.2 Identifiability with interventional data

In the previous section, we considered representation identification from some observational data where the data is generated from Equation (1). The latent variable $Z$ is drawn from an arbitrary distribution $\mathbb{P}_Z$, which does not necessarily come from any structural causal model. We next study how interventional data could enhance representation identification. We consider the general case where at least one component of $Z$, say the $i^{th}$ component, is set to a fixed value, and the remaining components are sampled from a distribution $\mathbb{Q}_{Z_{-i}}$. Note $\mathbb{Q}_{Z_{-i}}$ does not have to be equal to the distribution $\mathbb{P}_{Z_{-i}|z_i=z^*}$ over samples $Z_{-i}$ generated from Equation (1) when $z_i = z^*$. The data-generating process is written as follows

$$z_i \leftarrow z^*, \quad z_{-i} \sim \mathbb{Q}_{Z_{-i}}, \quad x \leftarrow g(z), \tag{5}$$

where the variable $z_i$ is fixed to be equal to $z^*$; the remaining $d - 1$ variables in $z$ (denoted as $z_{-i}$) are sampled from $\mathbb{Q}_{Z_{-i}}$; and the function $g$ is the true decoder that generates observed $x$ from $z$. The results we present below only require a restriction on the support of $\mathbb{Q}_{Z_{-i}}$ and this flexibility allows $\mathbb{Q}_{Z_{-i}}$ to model standard do interventions (Pearl, 2009), i.e., $Q_{Z_{-i}} = \mathbb{P}_{Z_{-i}|do(z_i=z^*)}$ as we illustrate below. Our DGP also allows the possibility that $\mathbb{Q}_{Z_{-i}} = \mathbb{P}_{Z_{-i}|z_i=z^*}$. We denote the support of $x$ under interventional data generated from equation (5) as $\mathcal{X}^i$. We write the reconstruction identity as follows.

$$h \circ f(x) = x, \forall x \in \mathcal{X} \cup \mathcal{X}^i \tag{6}$$

Further, we also enforce the constraint on the encoder that on all the data points $x \in \mathcal{X}^i$ one of the components (say $k$) takes some arbitrary fixed value $z^\dagger$. For all $x \in \mathcal{X}^i$.

$$f_k(x) = z^\dagger, \tag{7}$$

where $f_k(x)$ denotes the $k^{th}$ component which is required to take a fixed value $z^\dagger$.

**Structural causal model for $Z$ and associated interventions** In the work so far, we have not specifically invoked the condition that $Z$ is generated from a structural causal model. Our results are more general and apply to $Z$ that are not generated from a structural causal model as well. Now suppose $Z$ is drawn from a structural causal model. Say we observe both observational and interventional data. In the interventional data, in equation (5) exactly one latent undergoes a hard $do$

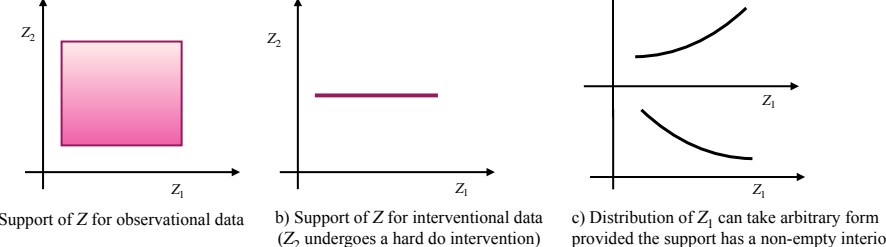

a) Support of $Z$ for observational data

b) Support of $Z$ for interventional data
($Z_2$ undergoes a hard do intervention)

c) Distribution of $Z_1$ can take arbitrary form
provided the support has a non-empty interior.

Figure 2: Illustration of assumptions on the support. In Figure 2a, we show that the support of both $Z_1, Z_2$ has a non-empty interior. In Figure 2b, hard $do$ intervention occurs on $Z_2$, the support of $Z_1$ has a non-empty interior. In Figure 2c, we show that for the setting corresponding to Figure 2b, the distribution $Z_1$ can take arbitrary form as long as the assumption on the support is met.

intervention. The constraint in equation (7) enforces that exactly one of the components of encoder also takes a fixed value. We illustrate this with a simple example. Suppose $Z = [Z_1, Z_2, Z_3, Z_4]$ is drawn from a structural causal model with the underlying DAG shown in Figure 1 a. Figure 1 b (c) shows the DAG when $Z_3(Z_2)$ undergoes a hard intervention and is set to $z_3^*(z_2^*)$. Under Figure 1 a), we show the reconstruction loss based on reconstruction identity. Under Figure 1 b,c, we show the reconstruction loss based on the reconstruction identity (equation (6)) along with a penalty that enforces the constraint in equation (7).

**Assumption 4.** *The interior of support of latents other than $i$, $\tilde{\mathcal{Z}}^i$, is a non-empty subset of $\mathbb{R}^{d-1}$.*

**Theorem 2.** *Suppose the observational data is generated from equation (1) and interventional data is generated from equation (5). If Assumption 1- 4 are satisfied, then the solution to equation (6) and (7) satisfy $\hat{z}_k = az_i + b$ and $\hat{z}_{-k} = Ez_{-i} + f$, where $\hat{z}_{-k}$ denotes the estimate of the latents other than $\hat{z}_k$ and $z_{-i}$ denotes the vector of true latents other than $z_i$, $a \in \mathbb{R}, b \in \mathbb{R}, E \in \mathbb{R}^{d-1 \times d-1}, f \in \mathbb{R}^{d-1}$, i.e., $z_i$ is identified up to shift and scaling, and $z_{-i}$ is affine identified.*

The proof of the above theorem is in the Appendix and we provide some intuition next. From Assumptions 1-3 hold, we can continue to use the result from previous Theorem 1 and claim affine identification. From affine identification it follows that $\hat{z}_k = a_{-i}^t z_{-i} + az_i + b$, where $z_{-i}$ is vector of $z$ other than $z_i$ and $a_{-i}$ is a vector of corresponding coefficients. Note that both $\hat{z}_k$ and $z_i$ are set to a fixed value. As a result, $a_{-i}^t z_{-i}$ takes a fixed value for all values of $z_{-i} \in \tilde{\mathcal{Z}}^i$. Suppose $a_{-i} \neq 0$ then this implies any changes to $z_{-i}$ in the direction of $a_{-i}$ will also reflect as a change $\hat{z}_k$ but that is a contradiction, which implies $a_{-i} = 0$. In Theorem 2, we did not make distributional assumptions (e.g., no parametric assumption) on $Z$ or on the nature of graphical model for $Z$ (e.g., $Z$ factorizes according to a certain DAG or a Markov random field). We only make assumptions on the support of $Z$ in observational data and interventional data. We require that the support of the variables that do not undergo a hard intervention have a non-empty interior. We illustrate this in Figure 2.

Suppose we have data from multiple environments, where each environment corresponds to a hard $do$ intervention on a distinct latent variable. Under same Assumptions as Theorem 2, we can identify each of the intervened latents up to permutation (since we do not know the index of the latent), shift, and scaling. We now consider a natural extension of the above setting. In the above setting, we assumed that in each environment at most one of the latent variables undergoes a hard $do$ intervention. Suppose multiple latent variables undergo a hard $do$ intervention. We can follow the exact proof recipe of Theorem 2 and achieve affine identification with respect to the block of hard intervened latents and block of remaining latents separately.

## 3  Conclusion

We studied the role of interventional data (level-two data) in causal representation learning. We show that latent variables that undergo $do$ interventions can be identified up to permutation, shift, and scaling when the true decoder is a finite degree polynomial. Extending these results beyond polynomials and $do$ interventions are important next steps.

# A  Appendix

We restate the theorems from the main body of the paper for convenience.

**Lemma 1.** *If the matrix $G$ that defines the polynomial $g$ is full rank, then $g$ is injective.*

**Proof** Suppose this is not the case and $g(z_1) = g(z_2)$ for some $z_1 \neq z_2$. Thus

$$
G \begin{bmatrix} 1 \\ z_1 \\ z_1 \bar\otimes z_1 \\ \vdots \\ \underbrace{z_1 \bar\otimes \cdots \bar\otimes z_1}_{p \text{ times}} \end{bmatrix} = G \begin{bmatrix} 1 \\ z_2 \\ z_2 \bar\otimes z_2 \\ \vdots \\ \underbrace{z_2 \bar\otimes \cdots \bar\otimes z_2}_{p \text{ times}} \end{bmatrix}
$$

$$
\implies G \begin{bmatrix} 0 \\ (z_1 - z_2) \\ z_1 \bar\otimes z_1 - z_2 \bar\otimes z_2 \\ \vdots \\ \underbrace{z_1 \bar\otimes \cdots \bar\otimes z_1}_{p \text{ times}} - \underbrace{z_2 \bar\otimes \cdots \bar\otimes z_2}_{p \text{ times}} \end{bmatrix} = 0 \tag{8}
$$

Since $z_1 \neq z_2$ we find a non-zero vector in the null space of $G$ which contradicts the fact that $G$ has full column rank. Therefore, it cannot be the case that $g(z_1) = g(z_2)$ for some $z_1 \neq z_2$. Thus $g$ has to be injective.

**Theorem 3.** *If the data generation follows equation* (1) *and Assumptions* 1-3 *hold, then the encoder $f$ and decoder $h$ that solve reconstruction identity in equation* (2) *achieve affine identification, i.e., $\forall z \in \mathcal{Z}, \hat{z} = Az + c$, where $\hat{z}$ is the output of the encoder $f$ and $z$ is the true latent and $A$ is an invertible $d \times d$ matrix.*

*Proof.* We start by restating the reconstruction identity. For all $x \in \mathcal{X}$

$$
h \circ f(x) = x
$$
$$
h(\hat{z}) = g(z)
$$
$$
H \begin{bmatrix} 1 \\ \hat{z} \\ \hat{z} \bar\otimes \hat{z} \\ \vdots \\ \underbrace{\hat{z} \bar\otimes \cdots \bar\otimes \hat{z}}_{p \text{ times}} \end{bmatrix} = G \begin{bmatrix} 1 \\ z \\ z \bar\otimes z \\ \vdots \\ \underbrace{z \bar\otimes \cdots \bar\otimes z}_{p \text{ times}} \end{bmatrix} \tag{9}
$$

Following the assumptions, $h$ is restricted to be polynomial but $f$ bears no restriction. If $H = G$ and $f = g^{-1}$, we get the ideal solution $\hat{z} = z$, thus a solution to the above identity exists. Since $H$ has full column rank, we can select $q$ rows of $H$ such that $\tilde{H} \in \mathbb{R}^{q \times q}$ and $\mathsf{rank}(\tilde{H}) = q$. Denote the corresponding matrix $G$ that select the same rows as $\tilde{G}$. We restate the identity in equation (9) in terms of $\tilde{H}$ and $\tilde{G}$ as follows.

$$\tilde{H} \begin{bmatrix} 1 \\ \hat{z} \\ \hat{z} \bar{\otimes} \hat{z} \\ \vdots \\ \underbrace{\hat{z} \bar{\otimes} \cdots \bar{\otimes} \hat{z}}_{p \text{ times}} \end{bmatrix} = \tilde{G} \begin{bmatrix} 1 \\ z \\ z \bar{\otimes} z \\ \vdots \\ \underbrace{z \bar{\otimes} \cdots \bar{\otimes} z}_{p \text{ times}} \end{bmatrix}$$

$$\begin{bmatrix} 1 \\ \hat{z} \\ \hat{z} \bar{\otimes} \hat{z} \\ \vdots \\ \underbrace{\hat{z} \bar{\otimes} \cdots \bar{\otimes} \hat{z}}_{p \text{ times}} \end{bmatrix} = \tilde{H}^{-1} \tilde{G} \begin{bmatrix} 1 \\ z \\ z \bar{\otimes} z \\ \vdots \\ \underbrace{z \bar{\otimes} \cdots \bar{\otimes} z}_{p \text{ times}} \end{bmatrix} \qquad (10)$$

$$\hat{z} = \tilde{A} \begin{bmatrix} 1 \\ z \\ z \bar{\otimes} z \\ \vdots \\ \underbrace{z \bar{\otimes} \cdots \bar{\otimes} z}_{p \text{ times}} \end{bmatrix}$$

$$\hat{z} = \tilde{A}_1 z + \tilde{A}_2 \ z \bar{\otimes} z + \cdots \tilde{A}_p \ \underbrace{z \bar{\otimes} \cdots \bar{\otimes} z}_{p \text{ times}} + c$$

Suppose at least one of $\tilde{A}_2, \cdots, \tilde{A}_p$ is non-zero. Among the matrices $\tilde{A}_2, \cdots, \tilde{A}_p$ which are non-zero, pick the matrix $\tilde{A}_k$ with largest index $k$. Suppose row $i$ of $\tilde{A}_k$ has some non-zero element. Now consider the element in the row in the LHS of (10) corresponding to $\hat{z}_i^p$. Observe that $\hat{z}_i^p$ is a polynomial of $z$ of degree $kp$, where $k \geq 2$. In the RHS, we have a polynomial of degree at most $p$. The equality between LHS and RHS is true for all $z \in \mathcal{Z}$ (and correspondingly all $x \in \mathcal{X}$). The difference of LHS and RHS is an analytic function. From Mityagin (2015) it follows that the LHS is equal to RHS on entire $\mathbb{R}^d$. If two polynomials are equal everywhere, then their respective coefficients have to be the same. Based on supposition, LHS has non zero coefficient for terms with degree $kp$ while RHS has zero coefficient for terms higher than degree $p$. This leads to a contradiction. As a result, none of $\tilde{A}_2, \cdots, \tilde{A}_p$ can be non-zero. Thus $\hat{z} = \tilde{A}_1 z + c$. Note that $\tilde{A}_1$ is also invertible. Suppose $\tilde{A}_1$ was not invertible, then we take a latent $z$ and perturb it in the direction in the null space of $\tilde{A}_1$. Note that under this perturbation $\hat{z}$ does not change but $z$ changes. Since $z$ changes $x$ has to change. However, $\hat{z}$ is same so the reconstructed $\hat{x}$ has to be the same. This leads to a violation of the reconstruction identity.

□

## A.1 Unknown degree of $g(\cdot)$

We provide further explanation for the case when we do not know the degree. The learner starts with solving the reconstruction identity by setting the degree of $h(\cdot)$ to be $s$; here we assume $H$ has full rank (this implicitly requires that $n$ is greater than the number of terms in polynomial of degree $s$).

$$H \begin{bmatrix} 1 \\ \hat{z} \\ \hat{z} \bar{\otimes} \hat{z} \\ \vdots \\ \underbrace{\hat{z} \bar{\otimes} \cdots \bar{\otimes} \hat{z}}_{s \text{ times}} \end{bmatrix} = G \begin{bmatrix} 1 \\ z \\ z \bar{\otimes} z \\ \vdots \\ \underbrace{z \bar{\otimes} \cdots \bar{\otimes} z}_{p \text{ times}} \end{bmatrix} \qquad (11)$$

We can restrict $H$ to rows such that it is a square invertible matrix $\tilde{H}$. Denote the corresponding restriction of $G$ as $\tilde{G}$. The equality is stated as follows.

$$
\begin{bmatrix}
1 \\
\hat{z} \\
\hat{z} \bar{\otimes} \hat{z} \\
\vdots \\
\underbrace{\hat{z} \bar{\otimes} \cdots \bar{\otimes} \hat{z}}_{s \text{ times}}
\end{bmatrix}
= \tilde{H}^{-1} \tilde{G}
\begin{bmatrix}
1 \\
z \\
z \bar{\otimes} z \\
\vdots \\
\underbrace{z \bar{\otimes} \cdots \bar{\otimes} z}_{p \text{ times}}
\end{bmatrix}
\tag{12}
$$

If $s > p$, then $\underbrace{\hat{z} \bar{\otimes} \cdots \bar{\otimes} \hat{z}}_{s \text{ times}}$ is a polynomial of degree at least $p + 1$. Since the RHS contains a polynomial of degree at most $p$ the two sides cannot be equal over a set of values of $z$ with positive Lebesgue measure in $\mathbb{R}^d$. Thus the reconstruction identity will only be satisfied when $s = p$. Thus we can start with the upper bound and reduce the degree of polynomial on LHS till the identity is satisfied.

## A.2 Extension from polynomials to $\epsilon$-approximate polynomials

We now discuss how to relax the polynomial assumption we discussed above. Suppose $g$ is a continuous function that can be $\epsilon$-approximated by a polynomial of degree $p$ on entire $\mathbb{R}^d$. If we continue to use $h$ as a polynomial, then satisfying the exact reconstruction is not possible. Instead, we enforce approximate reconstruction as follows. For all $x \in \mathcal{X}$ we want

$$
\| h \circ f(x) - x \| \le \epsilon
\tag{13}
$$

where $\epsilon$ is the tolerance on reconstruction error. We assume that $h(\cdot)$ is expressive enough that the above identity is satisfied up to $\epsilon$ tolerance. Recall $\hat{z} = f(x)$. We further simplify it as $\hat{z} = f \circ g(z) = a(z)$. We also assume that $a$ can be $\eta$-approximated on entire $\mathbb{R}^d$ with a polynomial of sufficiently high degree say $p$ and $q$ respectively. We write this as follows.

$$
\left\| \hat{z} - \Theta
\begin{bmatrix}
z \\
z \bar{\otimes} z \\
\vdots \\
\underbrace{z \bar{\otimes} \cdots \bar{\otimes} z}_{q \text{ times}}
\end{bmatrix}
\right\| \le \eta
$$

$$
\left\| \hat{z} - \Theta_1 z - \Theta_2 \, z \bar{\otimes} z - \cdots \Theta_p \, \underbrace{z \bar{\otimes} \cdots \bar{\otimes} z}_{q \text{ times}} \right\| \le \eta
\tag{14}
$$

We want to show that the norm of $\Theta_k$ for all $k \ge 2$ is sufficiently small. We state some assumptions needed in theorem below.

**Assumption 5.** *Encoder $f$ does not take values near zero, i.e., $f_i(x) \ge \gamma\eta$ for all $x \in \mathcal{X}$ and for all $i \in \{1, \cdots, d\}$, where $\gamma > 1$. The absolute value of each element in $\tilde{H}^{-1}\tilde{G}$ is bounded by a fixed constant. Consider the absolute value of the singular values of $\tilde{H}$; we assume that the smallest absolute value is strictly positive and bounded below by $\zeta$.*

**Theorem 4.** *Suppose $f : \mathbb{R}^n \to \mathbb{R}^d$ and $g : \mathbb{R}^d \to \mathbb{R}^n$ are functions such that $g$ and $a = f \circ g$ can be approximated by polynomials on entire $\mathbb{R}^d$. If $\mathcal{Z} = [-z_{\max}, z_{\max}]^d$, where $z_{\max}$ is sufficiently large, and Assumption 1,3, and 5 hold, then the polynomial approximation of $a$ (recall $\hat{z} = a(z)$) corresponding to solutions of approximate reconstruction identity in equation (13) is approximately linear, i.e., the norms of the weights on higher order terms is sufficiently small.*

**Proof sketch** We start by restating the approximate reconstruction identity. We use the fact that $g$ can be approximated with a polynomial of say degree $p$ to simplify the identity below.

$$\|h \circ f(x) - x\| \leq \epsilon$$

$$\left\| H \begin{bmatrix} \hat{z} \\ \hat{z} \bar{\otimes} \hat{z} \\ \vdots \\ \underbrace{\hat{z} \bar{\otimes} \cdots \bar{\otimes} \hat{z}}_{p \text{ times}} \end{bmatrix} - G \begin{bmatrix} z \\ z \bar{\otimes} z \\ \vdots \\ \underbrace{z \bar{\otimes} \cdots \bar{\otimes} z}_{p \text{ times}} \end{bmatrix} \right\| - \left\| G \begin{bmatrix} z \\ z \bar{\otimes} z \\ \vdots \\ \underbrace{z \bar{\otimes} \cdots \bar{\otimes} z}_{p \text{ times}} \end{bmatrix} - g(z) \right\| \leq \epsilon \tag{15}$$

Since $H$ is full rank, we select rows of $H$ such that $\tilde{H}$ is square and invertible. The corresponding selection for $G$ is denoted as $\tilde{G}$. We write the identity in terms of these matrices as follows.

$$\left\| \tilde{H} \begin{bmatrix} \hat{z} \\ \hat{z} \bar{\otimes} \hat{z} \\ \vdots \\ \underbrace{\hat{z} \bar{\otimes} \cdots \bar{\otimes} \hat{z}}_{p \text{ times}} \end{bmatrix} - \tilde{G} \begin{bmatrix} z \\ z \bar{\otimes} z \\ \vdots \\ \underbrace{z \bar{\otimes} \cdots \bar{\otimes} z}_{p \text{ times}} \end{bmatrix} \right\| \leq \frac{3\epsilon}{2}$$

$$\left\| \begin{bmatrix} \hat{z} \\ \hat{z} \bar{\otimes} \hat{z} \\ \vdots \\ \underbrace{\hat{z} \bar{\otimes} \cdots \bar{\otimes} \hat{z}}_{p \text{ times}} \end{bmatrix} - \tilde{H}^{-1} \tilde{G} \begin{bmatrix} z \\ z \bar{\otimes} z \\ \vdots \\ \underbrace{z \bar{\otimes} \cdots \bar{\otimes} z}_{p \text{ times}} \end{bmatrix} \right\| \leq \frac{3\epsilon}{2|\sigma_{\min}(\tilde{H})|} \tag{16}$$

where $|\sigma_{\min}(\tilde{H})|$ is the singular value with smallest absolute value corresponding to the matrix $\tilde{H}$. Now we write that the polynomial that approximates $\hat{z}_i = a_i(z)$ as follows.

$$\left| \hat{z}_i - \theta_1^{\mathsf{T}} z - \theta_2^{\mathsf{T}} z \bar{\otimes} z - \cdots \theta_q^{\mathsf{T}} \underbrace{z \bar{\otimes} \cdots \bar{\otimes} z}_{q \text{ times}} \right| \leq \eta \tag{17}$$

$$\hat{z}_i \geq \theta_1^{\mathsf{T}} z + \theta_2^{\mathsf{T}} z \bar{\otimes} z + \cdots \theta_q^{\mathsf{T}} \underbrace{z \bar{\otimes} \cdots \bar{\otimes} z}_{q \text{ times}} - \eta$$

$$\hat{z}_i \leq \theta_1^{\mathsf{T}} z + \theta_2^{\mathsf{T}} z \bar{\otimes} z + \cdots \theta_q^{\mathsf{T}} \underbrace{z \bar{\otimes} \cdots \bar{\otimes} z}_{q \text{ times}} + \eta \tag{18}$$

From Assumption 5 we know that $\hat{z}_i \geq \gamma \eta$, where $\gamma > 2$. It follows from the above equation that

$$\theta_1^{\mathsf{T}} z + \theta_2^{\mathsf{T}} z \bar{\otimes} z + \cdots + \theta_q^{\mathsf{T}} \underbrace{z \bar{\otimes} \cdots \bar{\otimes} z}_{q \text{ times}} - (\gamma - 1)\eta \geq 0 \tag{19}$$

For $\hat{z}_i \geq \gamma \eta$, let us track how $\hat{z}_i^p$ grows.

$$\hat{z}_i \geq \theta_1^{\mathsf{T}} z + \theta_2^{\mathsf{T}} z \bar{\otimes} z + \cdots \theta_q^{\mathsf{T}} \underbrace{z \bar{\otimes} \cdots \bar{\otimes} z}_{q \text{ times}} - \eta \geq 0$$

$$\hat{z}_i^p \geq (\theta_1^{\mathsf{T}} z + \theta_2^{\mathsf{T}} z \bar{\otimes} z + \cdots \theta_q^{\mathsf{T}} \underbrace{z \bar{\otimes} \cdots \bar{\otimes} z}_{q \text{ times}} - \eta)^p$$

$$\hat{z}_i^p \geq (\theta_1^{\mathsf{T}} z + \theta_2^{\mathsf{T}} z \bar{\otimes} z + \cdots \theta_q^{\mathsf{T}} \underbrace{z \bar{\otimes} \cdots \bar{\otimes} z}_{q \text{ times}})^p (1 - \frac{1}{\gamma - 1})^p \tag{20}$$

We consider $z = [z_{\max}, \cdots, z_{\max}]$. Consider of the terms $\theta_{ij} z_{\max}^k$ inside the polynomial in the RHS above. We assume all components of $\theta$ are positive. Suppose $\theta_{ij} \geq \frac{1}{z_{\max}^{k-\kappa-1}}$, where $\kappa \in (0, 1)$, then

the RHS grows at least $z_{\max}^{(1+\kappa)p}$. From equation (16), $\hat{z}_i^p$ is very close to degree $p$ polynomial in $z$. Under the assumption that the terms in $\tilde{H}^{-1}\tilde{G}$ are bounded by a constant the polynomial of degree $p$ grows at $z_{\max}^p$. The above expression in equation (20) grows at $z_{\max}^{(1+\kappa)p}\left(\frac{\gamma-2}{\gamma-1}\right)^p$. The difference in growth rates the equation (16) is an increasing function of $z_{\max}$ for ranges where $z_{\max}$ is sufficiently large. Therefore, the reconstruction identity in equation (16) cannot be satisfied for points in the neighborhood of $z = [z_{\max}, \cdots, z_{\max}]$. Therefore, $\theta_{ij} < \frac{1}{z_{\max}^{k-\kappa-1}}$. We can consider other vetries of the hypercube $\mathcal{Z}$ and conclude that $|\theta_{ij}| < \frac{1}{z_{\max}^{k-\kappa-1}}$

**Theorem 5.** *Suppose the observational data is generated from equation* (1) *and interventional data is generated from equation* (5). *If t Assumption* 1- 4 *are satisfied, then the solution to equation* (6) *and* (7) *satisfy* $\hat{z}_k = az_i + b$ *and* $\hat{z}_{-k} = Az_{-i} + c$, *where* $\hat{z}_{-k}$ *denotes the estimate of the latents other than* $k$ *and* $z_{-i}$ *denotes the vector of true latents other than* $i$.

*Proof.* First note that since Assumptions 1-3 hold, we can continue to use the result from Theorem 1. From Theorem 1, it follows that the estimated latents $\hat{z}$ are an affine function of the true $z$. $\hat{z}_i = a^t z + b, \forall z \in \mathcal{Z} \cup \mathcal{Z}^i$. From the constraint in equation (7) it follows that for all $z \in \mathcal{Z}^i$, $\hat{z}_i = z^\dagger$.

We write $z \in \mathcal{Z}^i$ as $[z^*, z_{-i}]$. We consider a $z \in \mathcal{Z}^i$ such that $z_{-i}$ is in the interior of $\tilde{\mathcal{Z}}^i$. We can write $\hat{z}_i = a_i z^* + a_{-i}^t z_{-i} + b$, where $a_{-i}$ is the vector of the values of coefficients in $a$ other than the coefficient of $i^{th}$ dimension, $a_i$ is $i^{th}$ component of $a$, $z_{-i}$ is the vector of values in $z$ other than $z_i$. We use these expressions to carry out the following simplification.

$$a_{-i}^t z_{-i} = z^\dagger - a_i z^* - b \tag{21}$$

Consider another data point $z' \in \mathcal{Z}^i$ from the same interventional distribution such that $z'_{-i} = z_{-i} + \theta e_j$, where $e_j$ is vector with one in $j^{th}$ coordinate and zero everywhere else. Since the point is from the same interventional distribution $z'_i = z^*$. From Assumption 4, we know that there exists a small enough $\theta$ such that $z'_{-i}$ is in $\mathcal{Z}^{-i}$ .

For $z'_{-i}$ we have

$$a_{-i}^t z'_{-i} = z^\dagger - az^* - b \tag{22}$$

We take a difference of the two equations (21) and (22) to get

$$a_{-i}^t(z_{-i} - z'_{-i}) = 0 \tag{23}$$

From the above we get that $a_j = 0$. We can repeat the above argument for all $j$ and get that $a_{-i} = 0$. $\qquad \square$

## B   Related Works

Our work relates to multiple threads of work in identifiable representation learning. We discuss them in groups based on the type of information they leverage for representation identification.

- **Time-series datasets.** Several works have leveraged the structure of latent variables in time-series data to achieve identification. The canonical data generating process in these works follows $x_t \leftarrow g(z_t)$ and the latents $z_t$ evolve under a structured time-series. Early works in this area consider non-stationary evolution of latents (i.e. assuming no dependence between time frames) (Hyvarinen and Morioka, 2016) and then came the models that considered stationary Markovian evolution (Hyvarinen and Morioka, 2017). In recent years, these models have been generalized significantly in works like Lachapelle et al. (2022); Ahuja et al. (2021); Lippe et al. (2022).

- **Contrastive observation-based datasets.** In another family of works including Zimmermann et al. (2021); Von Kügelgen et al. (2021); Brehmer et al. (2022); Locatello et al. (2020), one assumes access to contrastive observation pairs $(x, \tilde{x})$. For instance, an image and its rotated version can serve as contrastive observation pairs $(x, \tilde{x})$ (Zimmermann et al., 2021). In works such as Brehmer et al. (2022), the pair $(x, \tilde{x})$ corresponds to a data point pre and post-intervention on the latents. The data generation process is similar to time-series datasets in several aspects but there are a few key differences, including (a) the points $(x, \tilde{x})$ are not necessarily ordered by time, and (b) there may not exist any causal connections between the latents associated with $x$ and $\tilde{x}$, unlike those in time-series datasets (e.g., Lachapelle et al. (2022); Lippe et al. (2022)).

- **Auxiliary information datasets.** In the third line of work (Khemakhem et al., 2020a,b; Ahuja et al., 2022b), one assumes access to the high-dimensional observation $x$ (e.g., an image) and some auxiliary information $u$. If $u$ is the label of the image, we obtain standard supervised learning datasets. If $u = \tilde{x}$ is the positive pair (e.g., rotation of the image), we obtain contrastive observation-based datasets. If $u$ is the time stamp and previous time information $x_{t-1}$, then we obtain the time-series datasets. This line of work (Khemakhem et al., 2020a) often relies on strong assumptions on the interaction between latents and auxiliary information(e.g., latents are independent conditioned on auxiliary information) to guarantee provable identification.

Finally, we contrast our work with the existing works discussed above in terms of the type of information we leverage for representation identification: we leverage interventional knowledge (level-two knowledge in Pearl's ladder of causation (Bareinboim et al., 2022)) while most existing works leverage either observational data (level-one knowledge) (e.g. Khemakhem et al., 2020a) or counterfactual information (level-three knowledge) (e.g. Brehmer et al., 2022). Specifically in this work, we focus on studying "to what extent can we identify the latent causal variables if the data comprises different interventional distributions?" This question about causal representation learning shares the same spirit with causal discovery using interventional data, where we seek to understand how different interventional distributions help identify the underlying causal graph (Yang et al., 2018); both tasks rely on interventional data though targeting different causal tasks.

In contrast to our work, most existing works have leveraged other types of information for representation identification. For example, Brehmer et al. (2022) assume that the representation learner observes a pair of data points both pre- and post-intervention, and the data generation process therein requires the noise to be set to the same realization across the pair of points at all the nodes except the intervened nodes. Therefore, they leverage counterfactual information (level-three knowledge) for representation identification. As another set of examples, Lippe et al. (2022); Lachapelle et al. (2022) leverage pre- and post-intervention observations in adjacent time frames to study the causal relationships between the latents. Other works (e.g. Khemakhem et al. (2020a,b); Ahuja et al. (2022b)) directly work with observational data, i.e., level-one knowledge, and do not require or take advantage of interventional data. However, these works often achieve identification guarantees by making strong assumptions on the structure of the underlying causal connections between the latents, relying on observations of auxiliary information such as the label, or capitalizing on parametric assumptions on the distribution of the latent.

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
