# OpenReview forum: "Interventional Causal Representation Learning"
_NeurIPS.cc/2022/Workshop/nCSI — nCSI WS @ NeurIPS 2022 Oral_

### Official Review · Reviewer_YSJk · 2022-10-05
**Interesting new results under a new set of assumptions**

**Rating:** 3
**Confidence:** 3

**Review:**

Summary:
The authors do CRL from interventional data. They assume that the true and learned decoders are given by expanding the latent to all polynomial terms up to some order, followed by an injective linear map. They show that if an encoder exists, which can reconstruct all supported data points, the learned and true latent must be affinely related. Subsequently, when there exist interventional datasets in which one true latent does not change, but all other true latents do, this intervened latent variable can be identified by finding the constant dimension. Doing so, the learned latents differ from the true latents by a diagonal affine transformation.

Strengths:
- Strong conclusions from an an interesting new set of assumptions.
- In particular, the assumptions in theorem 2 seems quite applicable to real-world use-cases (once the latents have been affinely identified).
- Quite readable given the space constraints.

Weaknesses:
- A solid discussion about the applicability of the assumptions is missing.
- I'm not sure I agree with the fact that the full-rank properties are "minimal assumptions". Much work in causal representation learning is occupied in regimes in which the conclusion of theorem 1 is not satisfied.
- Regarding the generalisation beyond polynomial decoders: there is still a limitation that a general decoder can be approximated by a full-rank polynomial decoder. This excludes any non-full-rank polynomial decoder. The statement of theorem 4 is not very clear on this.

Suggestion for improvement/clarification:
- Say how one would actually learn the model. Currently, I only see a loss function in the caption of fig 1.
- I don't understand fig 2c.
- Clarify the definition of $\tilde A$ in (10) as the rows mapping to $\hat z$.
- In (4), replace $z$ with $\hat z$ to clarify that this concerns learned latents.
- Clarify: In Assumption 5, what is $\tilde G$ in the context of a non-linear g?

---

### Official Review · Reviewer_W7tg · 2022-10-15

**Rating:** 2
**Confidence:** 2

**Review:**

The authors study the identifiability of latent variables under observation of high-level variables that are generated via a polynomial function. The central result is identifiability of the latents up to affine functions under certain assumptions on the polynomial as well as support of the latents.

The paper is clear to read and the main ideas are conveyed quickly.

A major flaw with the writing is the notation of interventional data. In fact, the authors use $z_{-i} \sim \mathbb{P} (Z_{-i}\mid z_i = z^*)$, which does *not* denote the interventional data when $z_i$ is intervened via *do*, but simply the conditional probability after observing $z_i$ as a constant. The main outline of the paper as well as the motivation are thus misleading, even if it might have been a non intentional error (still a very major one). Upon initial reading, this mathematical mistake led me to question the correctness of the results. Nonetheless, the most important assumption seems to actually be the non-emptiness of the support of the latent variables (in particular Ass. 4) which can hold both for actual interventional data as well as observational data where we only observe a constant $z_i$.

While I haven’t checked the proofs in detail, on first glance they seem to be correct and rely on functional analysis, in particular Theorem 1.

I believe the paper would much benefit from even just a toy example to show how the latents can be learned, and it shouldn’t be too involved to code this down for a small example. As such, an important question and discussion for the results are the practicality — could a polynomial decoder give good empirical results on real-world data?

Since I cannot give more nuanced recommendations other than reject or accept, I decided to give a score of two as I believe the mistake can be fixed quickly and this is a workshop where we encourage a wide range of discussion, and I strongly advise the authors to correct the notation and make sure that the theoretical results still hold. Also, a discussion like the one mentioned above could be of value for the paper and discussions in the workshop.

---

### Meta-Review · Area_Chair_r858 · 2022-10-17

**Recommendation:** 2
**Confidence:** 4

**Metareview:**

The paper investigates the identifiability of latent causal representations in settings where the observed variables are polynomial transformations of the latent variables. Identifiability has not yet gotten enough attention in the causal representation learning literature and both reviewers agree that this work proposes an interesting perspective. However, reviewer W7tg has pointed to a crucial mistake in the definition of a do-intervention, which needs to be fixed. In summary, I recommend to accept this paper for the workshop conditioned on the authors fixing the mistake related to reviewer W7tg’s comment.

---

### Decision · Program_Chairs · 2022-10-20

Accept (Oral)